# Two-decade trends and factors associated with overweight and obesity among young adults in Nepal

Sujata Shakya[1]*, Pilvikki Absetz[2], Subas Neupane[2]

1 Central Department of Public Health, Tribhuvan University Institute of Medicine, Maharajgunj, Kathmandu, Bagmati Province, Nepal, 2 Unit of Health Sciences, Faculty of Social Sciences, Tampere University, Tampere, Finland

* sujata_8@iom.edu.np

**Data Availability Statement:** Datasets are freely accessible through Demographic and Health Survey Program, and are available to anyone on request at https://dhsprogram.com/methodology/survey-search.cfm?sendsearch=1&sur_status=

## Abstract

Overweight and obesity are global epidemics in the adult population, and also affect young people. This study estimated the long-term trend (1996–2019) of overweight and obesity among young adults aged 18 to 29 years in Nepal by sex based on the World Health Organization (WHO) and Asian cut-offs for body mass index (BMI). We also investigated the demographic factors associated with overweight and obesity in the latest survey. This study utilized data from nationwide studies, Demographic and Health Surveys (DHSs) and WHO STEPwise approach to surveillance (STEPS) surveys. The trends in overweight and obesity were studied using trend analysis and joinpoint regression. Average annual percent changes (AAPCs) and their 95% confidence intervals (CIs) were calculated for the trends. Multivariable logistic regression was used to study the factors associated with overweight and obesity. The study findings showed significant upward trends in both overweight and obesity for women with AAPCs of 10.5 (95% CI 6.4–14.7) and 15.8 (95% CI 10.9–20.8) respectively. In the 25–29 age-group, the prevalence of overweight/obesity increased among women from 2.2% to 24.7% between 1996 and 2019, and among men from 8.8% to 25.4% between 2007 and 2019. Increased odds of overweight (AOR 9.15, 95% CI 6.64–12.60), and obesity (AOR 42.09, 95% CI 10.12–175.04) were found in 2019 compared to 1996. Older age and female sex, being married and urban residence were significantly associated with overweight and obesity. In conclusion, this study showed rapid upward trends in overweight and obesity among young adults in Nepal with an accelerated trend among women; the predictors for overweight were older age, female sex and married status, and those for obesity were older age, female sex, and urban residence.

## Introduction

Overweight and obesity are global public health problems, caused by accumulation of excessive body weight and fat. This is commonly measured by BMI and other measures such as the waist-to-hip ratio and waist-to-height ratio [1–3]. The WHO has characterized the increasing prevalence of obesity as a 'global epidemic' and an imminent global threat [3, 4]. In 2014, 11%

Completed&str1=13,&str2=1,2,3,17,&crt=1&listgrp=0.

**Funding:** The authors received no specific funding for this work.

**Competing interests:** The authors have declared that no competing interests exist.

of men and 15% of women aged 18 years and above were obese [5]. Obesity in all age groups is the fifth most common cause of the disease burden, ranking just below underweight in middle-income countries in Eastern Europe, Latin America, and Asia [6]. The global obesity prevalence has nearly doubled since 1980 [5] and tripled in low- and middle-income countries (LMICs) over the past three decades due to urbanization and economic transition [6, 7], in addition to a high prevalence of undernutrition [8]. By 2030, globally, 2.16 billion and 1.12 billion people are expected to be overweight and obese, respectively [7]. Overweight and obesity account for 16% of the global burden of disease in terms of disability-adjusted life years [9]. A high BMI accounts for 28.6 million years lived with disability [10]. Obesity is directly related to health risks and premature deaths from chronic debilitating disorders including cardiovascular and respiratory diseases, diabetes, osteoarthritis, infertility and certain types of cancer [6, 11–13]. Thus, it is a major burden on individuals and health systems.

The dominant factors of obesity include concurrent rapid changes in diet (global nutrition transition), such as al fresco dining, snacks between meals and less physical activity [6, 7, 14]. Heavy consumption of alcohol, cigarettes and meat and a sedentary lifestyle are associated with a significantly higher BMI [15]. Among sociodemographic factors, a strong inverse relationship was depicted between educational status and NCDs in Central and Eastern Europe [16]. Another study found that male sex, high socioeconomic status, sedentary behaviour, and less consumption of fruits were associated with overweight in Nepal [17]. Overweight-obesity was found to be more prevalent among urban residents in Bangladesh, India and Nepal [18]. However, overweight and obesity in most LMICs is increasing in both rural and urban areas and has even sharply increased in rural areas [16].

Obesity, which was once considered to be most common among middle aged adults, is now also becoming prevalent among adolescents and young adults [19]. Young adulthood from age 18 to 25 is a critical stage in a person's life course [20], during which large lifestyle changes occur such as leaving home, starting a career, and experiencing pregnancy and child rearing. These factors create challenges and stress, making young adults vulnerable to weight gain [21–24]. Moreover, young adults generally give less priority to their current and future health. Any type of health behaviour established during this transition often persists later in life. The definition of young adulthood is still unclear, and the age range varies depending upon different studies [20]. A systematic review conducted in different LMICs included the age range of 16–30 years, considering this the transitional age for most global social settings [20]. Another study reported the fastest increase in overweight and obesity prevalence among individuals aged 18–30 years [25].

The prevalence of obesity ranges from 2.3 to 12% among young adults in LMICs, and the prevalence of overweight is 28.8% [20]. Overall, 22% of young adults in 22 LMICs in 2014 were overweight/obese. Different cross-sectional studies in Nepal showed a high prevalence in the young adult group [15, 17, 26]. This is alarming for a low-income country such as Nepal, where the problem of undernutrition has not yet been resolved.

There is inadequate evidence in Nepal to explain the current trends of overweight and obesity among young adults and the socioeconomic factors that affect them. There are few findings from small-scale cross-sectional studies [15, 17, 18, 26], through which it is challenging to draw concrete conclusions regarding the current trends. In addition to the researcher's knowledge, long-term trends have not been studied thus far. Taking lessons from other LMICs, there is an urgent need to trace the current situation of overweight and obesity among young adults, so that appropriate country-specific interventions can be planned to mitigate this situation. Therefore, this study based on two national-level surveys, aimed to analyse the long-term trends (1996–2019) of overweight and obesity among the young adults in Nepal based on the WHO BMI classification [27] and Asian cut-offs [12]. BMI was supplemented by the waist-to-

hip ratio and waist-to-height ratio for WHO surveys, and the sociodemographic factors associated with overweight and obesity. The information on twenty year long-term trends can provide policy-makers with information on the current situation, gaps and risk factors which can help them in planning future preventive strategies.

## Materials and methods

We used data from several cross-sectional surveys conducted among nationally representative adult samples from 1996–2019. Two existing data sources, namely, the Demographic and Health Surveys (DHSs), and the WHO STEPwise approach to NCD risk factor surveillance (STEPS) of Nepal were used. We used data only from young adults since our concern was overweight and obesity among this group. We considered young adults aged 18–29 years, similar to a previous study conducted in Nepal [28]. The total number of participants aged 18–29 years enrolled in the DHS and WHO STEPS surveys in different years was 21,281.

### Demographic and Health Surveys (DHSs)

The DHSs are cross-sectional surveys conducted every 5 years among nationally representative samples. We utilized data on overweight and obesity based on BMI and related independent variables from five consecutive surveys between 1996 and 2016 conducted under the Ministry of Health and Population (MoHP), Nepal. The objective of the surveys was to provide up-to-date estimates of basic demographic and health indicators. The DHSs adopted multistage stratified cluster sampling techniques by using probability proportional to size of population (PPS). In total, approximately 30,000–40,000 adults aged 15–49 years were interviewed in all surveys. Data for the survey were collected through house-to-house structured interviews. The DHSs generated high-quality data on important demographic, economic, social and health factors for LMICs and have been used in high-quality research [29].

### WHO STEPS surveys

The WHO STEPS surveys were conducted under the supervision of the MoHP, Nepal, to determine the prevalence of modifiable behavioural risk factors for NCDs over time. The STEPS surveys are population-based surveys among adults aged 15–69 years conducted using internationally comparable, standardized and integrated surveillance tools. The surveys were conducted nationwide in 2007, 2012–13 and 2019. The survey adopted multistage stratified cluster sampling techniques and the data collection was performed through house-to-house interviews. The overall sample size taken for WHO STEPS surveys was on average 4,000–5,000 [30, 31]. From these surveys, we accessed data on overweight and obesity based on BMI, waist-to-hip ratio and waist-to-height ratio, and other sociodemographic factors.

### Ethical considerations

The DHS and WHO STEPS surveys were approved by the Nepal Health Research Council (NHRC) Ethical Review Board. Informed consent for the surveys was obtained from all the participants who were interviewed [29–31]. In this study, a separate approval from the ethical board was not required. The confidentiality and anonymity of the participants were maintained in all the surveys. There are no names of individuals, household addresses or any identifiable data in the data files. Permission for accessing the data was granted by the DHS program and WHO NCD microdata repository.

### Measurement of variables

**Overweight and obesity.** We used BMI to define overweight and obesity. Both the WHO universal classification (overweight: BMI $\geq$25 and <30.0 kg/m$^2$, obesity: BMI $\geq$30.0 kg/m$^2$) [27] and the WHO Asian classification (overweight: BMI of 23.0–27.5 kg/m$^2$, obesity: BMI>27.5 kg/m$^2$) [12] were used to classify participants as overweight or obese. The waist-to-hip ratio and waist-to-height ratio were calculated only from the data of the WHO STEPS surveys and these are presented in the S2 Fig. We included these measures since obese individuals differ not only in the amount of body fat deposits but also in the regional distribution of fat [1–3]. BMI information for men was available from the STEP surveys and only one of the DHSs, therefore the trend analysis for men was limited to between 2007 and 2019.

**Sociodemographic characteristics and other independent variables.** Sociodemographic information such as age in years (<20, 20–24, 25–29), sex (male, female), marital status (never married, married, widowed/divorced), religion (Hindu, Buddhist, Muslim, Christian), ethnicity (Brahmin/Chhetri, Janajati, Madhesi, Dalit, Muslim, other), place of residence (urban, rural), region (eastern, central, western, mid-western, far-western), ecological zone (mountain, hill, Terai), educational level (no education/preschool, primary, secondary, higher) and occupation (professional/technical, managerial, clerical, sales/services, agriculture, manual/labour, did not work) were extracted from the survey data. Similarly, information on the use of contraceptives (hormonal, nonhormonal, not used) and overall tobacco use, including both smoking and smokeless tobacco (yes, no) were used in the analysis. For regression analysis, some of the variables were recategorized by collapsing small categories such as marital status, which was dichotomized as never married vs. married/widowed/divorced.

### Statistical analysis

The national survey data are publicly available to use with permission from the authorities [32]. The data analysis was performed in SPSS version 16 (IBM Corp., Chicago, IL) and Joinpoint version 4.9.1.0 (IMS, Inc.). The data were weighted to make them representative of the study population and to adjust for variations in the probability of sample selection. We explored missing values in the data before performing the analysis. There were few missing values in the whole datasets, and the missing values were imputed when applicable before data were released for public use in the DHSs and WHO STEPS surveys.

The results were summarized using frequencies, percentages, means/medians and standard deviations based on the nature of the variables and their distribution. Joinpoint regression and multiple logistic regression analyses were used to analyse the temporal trends of overweight and obesity. For the trend analysis using joinpoint regression, we used a permutation test to select the final model that defined the best-fitting regression line and the number of joinpoints to assess significant changes in the trends. This approach uses a sequence of permutation tests to decide the best number of joinpoints [33]. There were 4500 permutations and the final model was selected based on the annual percent changes (APCs) between points of change in trends that were significantly different from zero at the alpha 0.05 level. Statistical significance was tested starting from 0 to 1 joinpoint. The joinpoints were restricted to 0 and 1 joinpoints since we only had 8 observation points. We obtained the AAPC, which is the weighted average of the changes over a specific period [34]. The APC and AAPC are equal for zero joinpoint when there is no change in trend. We could not perform joinpoint regression for men since there were only 4 measurement points. The trends in overweight and obesity based on WHO and Asian cut-offs are presented separately for men and women in the S2 Fig, and the median waist-to-hip ratio and waist-to-height ratio were calculated from the 2007 to 2019 WHO surveys. We assessed the age-specific trends of overweight-obesity in the three age groups and

present them in graphs for men and women. The interaction effect of age group and sex on overweight and obesity was examined in a regression model. To study the predictive power of the age groups, we calculated the predictive margins of the age groups and their 95% CIs by sex for overweight and obesity.

We studied the association of sociodemographic factors with overweight and obesity based on the latest DHS from 2016, since this was the most recent survey that included information on all sociodemographic factors. Logistic regression models were used to study the association of overweight and obesity with sociodemographic factors. Crude and multivariable models were fitted. Variables that were significant at the 10% significance level in the crude model were used in the multivariable model. Possible multicollinearity between variables was tested using the variance inflation factor (VIF) and tolerance index (TI). Variables with VIF values greater than 5 and TI values less than 0.2 were regarded as having high multicollinearity and thus were excluded from the final model [35]. The outcomes of the regression analysis were presented as odds ratios and their 95% confidence intervals.

## Results

### Demographic characteristics

The demographic characteristics of the respondents in different years from 1996 to 2019 are illustrated in Table 1. The mean age of the respondents was approximately the same in all surveys, with the lowest mean age in 2016 (22.98±3.47) and the highest in 2012 (24.15±3.31). All the DHSs excluding that in 2016 included only women, whereas the WHO STEPS surveys included both men and women. In most of these surveys, women were the majority. The majority of the respondents were rural residents, except for those in the year 2016 survey, in which nearly two-thirds were from urban areas. The proportion of the study population with a higher educational level increased from earlier to recent surveys, and the proportion of those with a primary level of education or below decreased in recent surveys. Separate tables for men and women (S1 and S2 Tables) showed a higher proportion of men as urban dwellers and having higher education levels.

### Trends in overweight and obesity

Fig 1 depicts the trends in overweight and obesity among women by using joinpoint regression. Fig 1A shows a significant and steady increase in the prevalence of overweight from 1996 to 2019 with an average annual percent change (AAPC) of 10.5. We identified two periods/segments (1996–2012 and 2012–2019) in the trend of overweight among women (Fig 1B). At one joinpoint (Fig 1B), the annual percent change (APC) for the first segment was 12.0, which was significantly different from zero, while that of the latter segment was not significant (APC = 6.3). The trend of obesity showed a similar and significant increase over the period, with an AAPC of 15.8 (Fig 1C), whereas with 1 joinpoint, only the APC of the first segment was significant (Fig 1D). The final selected models for both overweight and obesity were those with a 0 joinpoint where the APC equaled the AAPC.

We present the trends of overweight and obesity prevalence in S1 Fig, which shows that the prevalence of overweight among women increased steadily from 1996 (1.5%) to 2019 (14.6%), with a slight drop in 2007 and 2016 (S1 Fig). Similarly, a gradual rise in the trends of obesity over the same period was found, with a steady rise from 1.2% in 2011 to 3.3% in 2019. The prevalence of overweight and obesity with the Asian cut-offs was higher than that with the WHO universal cut-offs. The median waist-to-hip ratio and waist-to-height ratio were calculated based on WHO data (S2 Fig), which shows a gradual increase for both men and women

**Table 1. Demographic characteristics of the participants by survey year.**

| Background characteristics | Years of survey | | | | | | | |
|---|---|---|---|---|---|---|---|---|
| | **1996** | **2001** | **2006** | **2007** | **2011** | **2012** | **2016** | **2019** |
| | n = 3305 (%) | n = 3352 (%) | n = 4203 (%) | n = 1195[†] (%[‡]) | n = 2492 (%) | n = 835[†] (%[‡]) | n = 3992 (%) | n = 1196[†] (%[‡]) |
| **Age (in years)** | | | | | | | | |
| < 20 | 447 (13.5) | 424 (12.7) | 826 (19.7) | 220 (21.5) | 468 (18.8) | 89 (12.3) | 834 (20.9) | 127 (13.5) |
| 20–24 | 1412 (42.7) | 1453 (43.3) | 1794 (42.7) | 497 (48.9) | 1079 (43.3) | 316 (41.1) | 1689 (42.3) | 462 (37.9) |
| 25–29 | 1446 (43.8) | 1475 (44.0) | 1538 (37.6) | 478 (29.6) | 945 (37.9) | 430 (46.6) | 1469 (36.8) | 607 (48.7) |
| Mean age | 23.70 (±3.32) | 23.69 (±3.30) | 23.18 (±3.44) | 23.24 (±3.46) | 23.20 (±3.45) | 24.15 (±3.31) | 22.98 (±3.47) | 24.11 (±3.23) |
| **Sex** | | | | | | | | |
| Male | - | - | - | 541 (54.1) | - | 239 (47.1) | 1525 (38.2) | 362 (46.0) |
| Female | 3305 (100) | 3352 (100) | 4203 (100) | 654 (45.9) | 2492 (100) | 596 (51.7) | 2467 (61.8) | 834 (54.0) |
| **Residence** | | | | | | | | |
| Urban | 291 (8.8) | 323 (9.6) | 724 (17.2) | 583 (19.5) | 378 (15.2) | 158 (20.4) | 2575 (64.5) | 141 (8.9) |
| Rural | 3014 (91.2) | 3028 (90.4) | 3479 (82.8) | 612 (80.5) | 2114 (84.8) | 677 (79.6) | 1417 (35.5) | 1055 (91.1) |
| **Educational level** | | | | | | | | |
| No education/preschool | 2369 (71.7) | 2122 (63.3) | 1677 (39.9) | 222 (22.7) | 672 (27.0) | 243 (21.3) | 506 (12.7) | 265 (19.6) |
| Primary | 457 (13.8) | 557 (16.6) | 839 (20.0) | 391 (40.7) | 474 (19.0) | 135 (16.4) | 640 (16.0) | 255 (21.0) |
| Secondary | 411 (12.4) | 612 (18.2) | 1345 (32.0) | 253 (21.3) | 1018 (40.8) | 238 (31.6) | 1684 (42.2) | 379 (34.2) |
| Higher | 69 (2.1) | 60 (1.8) | 342 (8.1) | 328 (15.3) | 328 (13.2) | 219 (30.7) | 1162 (29.1) | 297 (25.2) |

[†]unweighted frequencies; weighted total population size for 2007 survey = 2122829, for 2012 = 6241140, and for 2019 = 6240776

[‡]weighted percentage

and an even faster increase for women, although a lower median value (0.54) in 2007 increased to 0.87 in 2019.

Table 2 shows the details of APC and AAPC with 0 and 1 joinpoint for both overweight and obesity among women. The AAPCs for 0 joinpoints were significant whereas only the AAPC of obesity for 1 joinpoint was significantly different from zero.

## Age-specific trends of overweight-obesity

The age-specific prevalence of overweight-obesity among women showed an upward trend for all age-groups (Fig 2B). Overweight-obesity prevalence as lowest in the youngest cohort (<20 years), although it showed a slow upward trend from 0% in 1996 to 10.4% in 2019. However, there was a sharp increase in the trend in the oldest cohort (25–29 years), from 2.2% in 1996 to 24.7% in 2019, the highest among all age groups. There was a slight drop during 2007.

The prevalence of overweight-obesity among men also showed gradual rise from 2007 to 2019 with slight decrease in 2016 (Fig 2A). The prevalence of overweight-obesity for all the years was highest in the 25–29 age group, which increased from 8.8% in 2007 to 25.4% in 2019.

A descriptive analysis showing trends of overweight-obesity among women based on area of residence is presented in the S1 Fig. The analysis showed that the highest proportions of overweight and obesity among women were among urban dwellers, and this showed a rapid upward trend from 1996 to 2019.

Table 3 shows the change in overweight and obesity and OR with 95% CIs. Compared to the year 1996, more than twofold increased odds of overweight were found in 2001, which increased to approximately ninefold in 2019. The odds of an increase in the obesity prevalence were even higher, from approximately 6-fold in 2001 to 42-fold in 2019. The change in overweight and obesity based on Asian cut-offs is presented in S1B Fig, which shows a similar

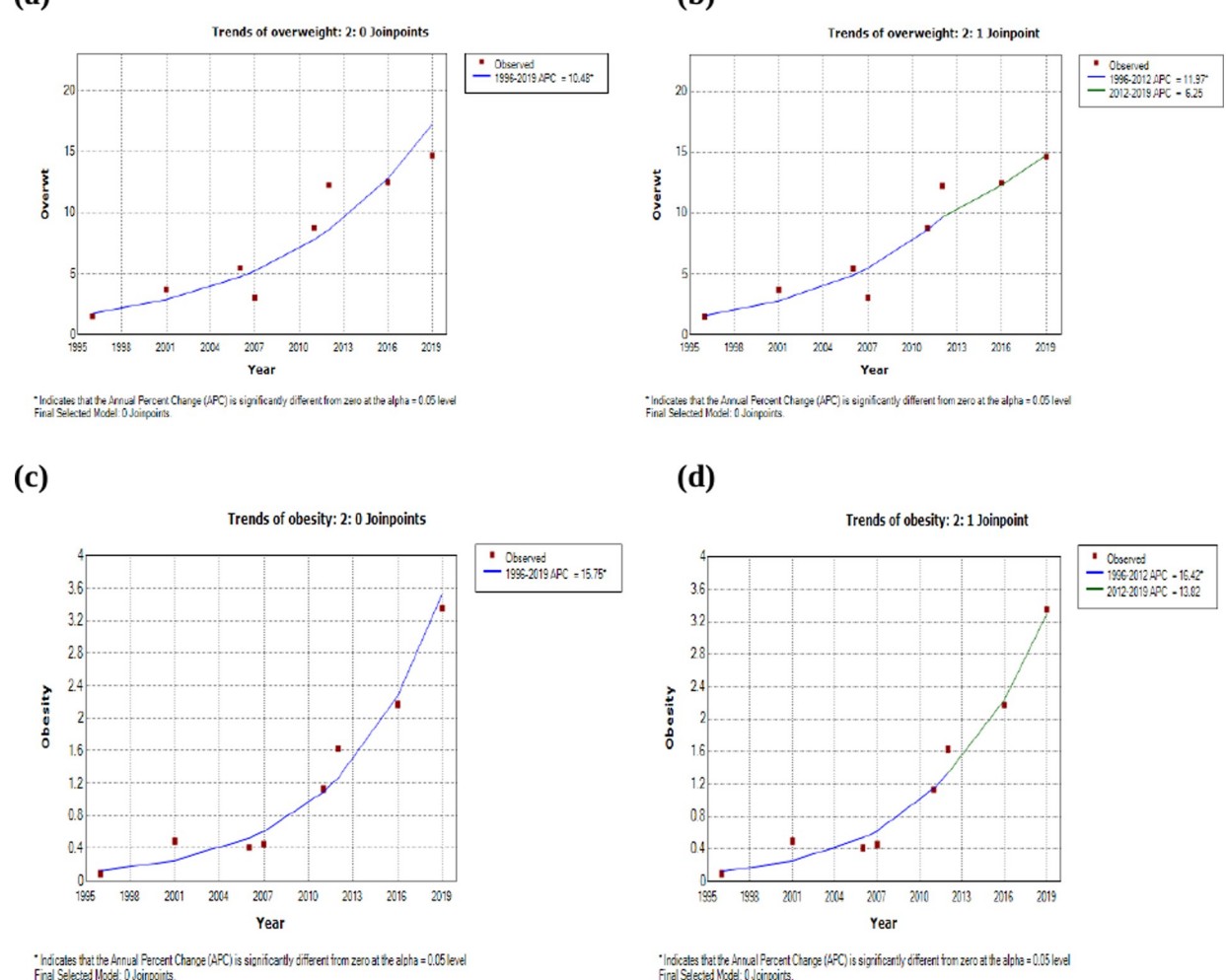

**Fig 1.** **(a)** Joinpoint regression analysis of overweight in women (0 joinpoint). **(b)** Joinpoint regression analysis of overweight in women (1 joinpoint). **(c)** Joinpoint regression analysis of obesity in women (0 joinpoint). **(d)** Joinpoint regression analysis of obesity in women (1 joinpoint).

increase, that is, approximately twice increased the odds of overweight in 2001 to 4 times in 2019, whereas, obesity increased by odds of approximately 4 times in 2001 to 28 times in 2019.

The unadjusted model (Model 1) in Table 4 shows that age, sex and area of residence were significantly associated with both overweight and obesity, whereas marital status was

**Table 2. Joinpoint regression analysis of overweight and obesity among young adult women in Nepal.**

| Outcomes | 0 joinpoint | | 1 joinpoint | | |
|---|---|---|---|---|---|
| | Period | AAPC (95% CI) | Period | APC (95% CI) | AAPC (95%CI) |
| Overweight | 1996–2019 | 10.5 (6.4–14.7)* | 1996–2012 | 12.0 (1.2–23.9)* | 10.2 (-1.5–23.3) |
| | | | 2012–2019 | 6.3 (-38.9–84.7) | |
| Obesity | 1996–2019 | 15.8 (10.9–20.8)* | 1996–2012 | 16.4 (2.2–32.6)* | 15.6 (0.1–33.6)* |
| | | | 2012–2019 | 13.8 (-44.1–131.7) | |

APC: Annual Percent Change, AAPC: Average Annual Percent Change, CI: Confidence Interval

*statistically significant at 0.05

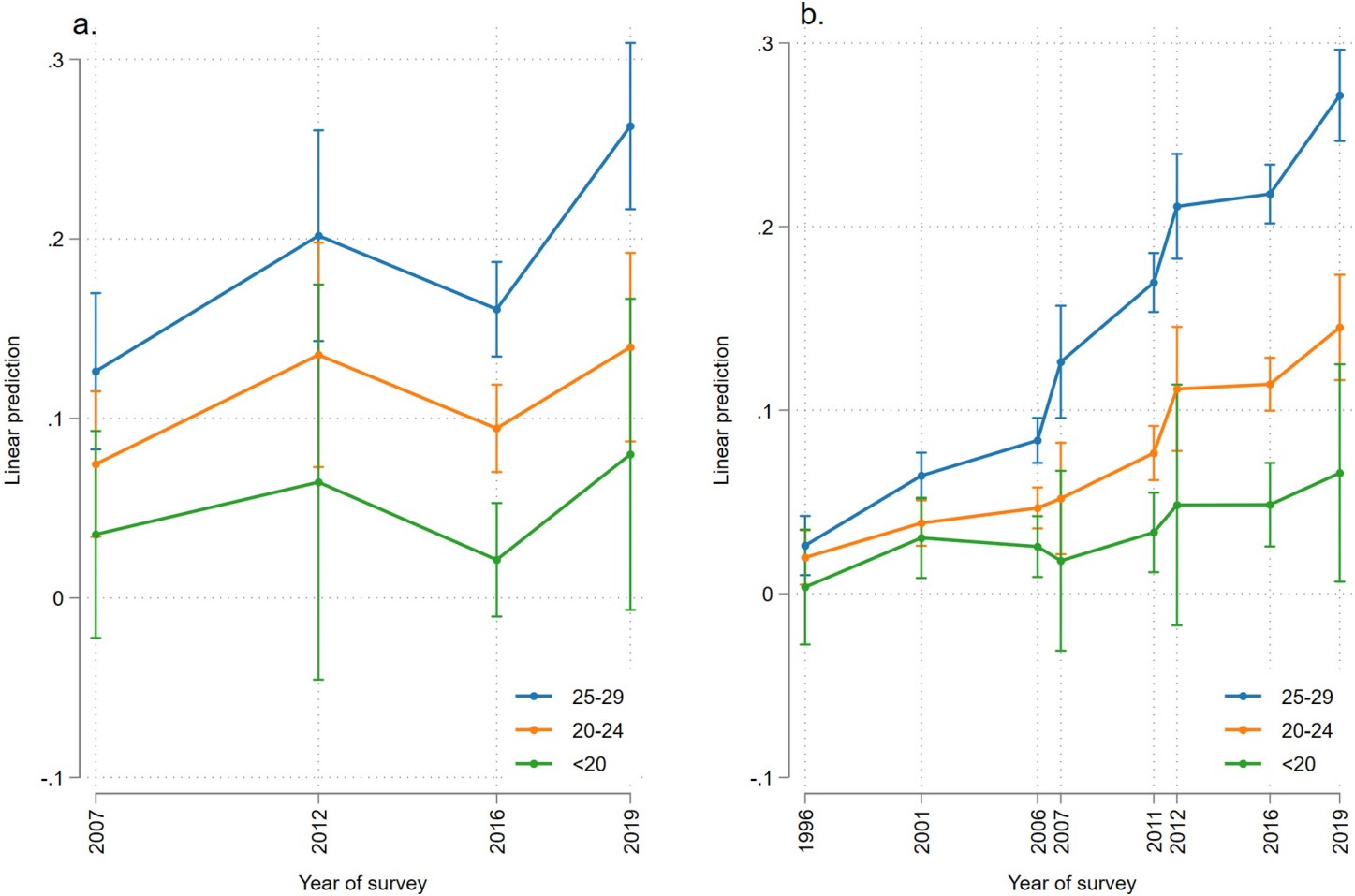

**Fig 2. (a)** Age-specific trends in the prevalence of overweight-obesity among men of 18–29 years from 2007 to 2019 using the WHO cut-off **(b)** Age-specific trends in the prevalence of overweight-obesity among women aged 18–29 years from 1996 to 2019 using the WHO cut-off.

associated with only overweight. In the adjusted multivariable model (Model 2), age, sex and marital status were strong predictors for overweight, while age, sex and area of residence were predictors for obesity. Compared to the youngest age group, young adults aged 20–24 years

**Table 3. Change in overweight and obesity based on the survey year and odds ratios (ORs) with their 95% CIs (WHO BMI categories).**

| Years of Survey | Number | Overweight | Obesity |
|---|---|---|---|
| | | OR (95% CI) | OR (95% CI) |
| **1996** | 2511 | Ref | Ref |
| **2001** | 3991 | 2.17 (1.58–2.99) | 5.93 (1.38–25.50) |
| **2006** | 4881 | 2.74 (2.01–3.72) | 3.70 (0.84–16.29) |
| **2007** | 1224 | 4.17 (2.92–5.96) | 11.72(2.56–53.59) |
| **2011** | 2826 | 4.54 (3.33–6.19) | 18.31 (4.42–75.87) |
| **2012** | 858 | 7.26 (5.15–10.23) | 26.16 (6.03–113.53) |
| **2016** | 4384 | 5.87 (4.36–7.90) | 21.66 (5.31–88.42) |
| **2019** | 1192 | 9.15 (6.64–12.60) | 42.09 (10.12–175.04) |

**Table 4. Association of sociodemographic factors with overweight and obesity in 2016 (based on WHO criteria).**

| Factors | Number | Overweight | | Obesity | |
|---|---|---|---|---|---|
| | | Crude OR (95% CI) | AOR[†] (95% CI) | Crude OR (95% CI) | AOR[†] (95% CI) |
| **Age group (in years)** | | | | | |
| Less than 20 | 855 | Ref | Ref | Ref | Ref |
| 20–24 | 1851 | 2.84 (1.90–4.25) | 2.09 (1.37–3.20) | 3.65 (1.28–10.37) | 4.14 (1.23–13.92) |
| 25–29 | 1519 | 5.81 (3.92–8.63) | 3.79 (2.47–5.81) | 5.48 (1.94–15.46) | 6.20 (1.80–21.34) |
| **Sex** | | | | | |
| Male | 1485 | 0.69 (0.56–0.85) | 0.68 (0.52–0.87) | 0.54 (0.31–0.94) | 0.43 (0.23–0.80) |
| Female | 2740 | Ref | Ref | Ref | Ref |
| **Marital status** | | | | | |
| Never married | 1296 | Ref | Ref | Ref | Ref |
| Married/widow/divorced | 2929 | 2.76 (2.12–3.59) | 2.03 (1.50–2.76) | 1.61 (0.92–2.83) | 1.11 (0.59–2.10) |
| **Place of residence** | | | | | |
| Urban | 2774 | 1.29 (1.04–1.59) | 1.23 (0.99–1.54) | 2.70 (1.45–5.05) | 2.36 (1.25–4.45) |
| Rural | 1451 | Ref | Ref | Ref | Ref |
| **Occupation** | | | | | |
| Prof/tech/managerial | 226 | 1.39 (0.93–2.07) | 1.43 (0.94–2.17) | 0.72 (0.21–2.45) | 0.72 (0.21–2.50) |
| Clerical | 119 | 1.17 (0.66–2.06) | 1.40 (0.77–2.55) | 0.96 (0.22–4.17) | 1.18 (0.26–5.33) |
| Sales/services | 508 | 1.53 (1.13–2.07) | 1.55 (1.12–2.15) | 2.29 (1.21–4.34) | 2.47 (1.26–4.83) |
| Agriculture | 1629 | 0.68 (0.53–0.88) | 0.63 (0.49–0.82) | 0.53 (0.27–1.02) | 0.54 (0.28–1.05) |
| Manual/labour | 560 | 0.88 (0.63–1.24) | 0.98 (0.67–1.42) | 1.03 (0.48–2.23) | 1.38 (0.61–3.16) |
| Did not work | 1183 | Ref | Ref | Ref | Ref |

[†]Adjusted for age, sex, residence, occupation and marital status

and 25–29 years had increased odds of being overweight. The magnitude of the association was stronger for the oldest age group (OR 3.79, 95% CI 2.47–5.81). Similarly regarding obesity, the younger group (20–24 years) had approximately four times increased odds, and the oldest group (25–29 years) had six times higher odds of being obese. Males had 32% and 57% lower odds of being overweight and obese, respectively, than their female counterparts. Similarly, the married population had twice the odds of being overweight compared to the never-married population. Urban residents had twice the odds of being obese compared to rural residents.

## Discussion

Our study showed sharp upward trends in overweight and obesity among Nepalese young adults, especially among women. The prevalence of overweight among women showed a rapid increase from 1996 to 2019. The rise in the trend of obesity among women was even faster. The overweight-obesity prevalence was highest in the 25–29 age among both women and men, with a relatively rapid upward trend among women. The likelihood of overweight was increased in groups of older age, female sex and married status, whereas the likelihood of obesity was increased in groups of older age, female sex and urban residence.

We found very limited studies on trends of overweight/obesity prevalence, especially those focusing young adults for comparison with our findings. Due to differences in the methodology in other studies, we found only a few studies comparable with ours.

### Prevalence and trends of overweight and obesity

The latest survey among a nationally representative sample of young Nepalese adults, included in the present study, showed an overweight-obesity prevalence for both sexes below 20%.

Among women, these findings are comparable to those in the study by Peltzer et. al study in 22 LMICs (19.3%), while among men, the prevalence in the 22 LMICs was considerably higher (24.%) [36]. All these figures showed higher proportions of overweight or obesity among men.

Our study showed a sharp increase in the trend in overweight among women, with an AAPC of 10.5 (p<0.05). A slight change in the trend was noticeable in 2012; however, the AAPC was not significant after 2012. The prevalence of obesity was much lower than that of overweight; nevertheless, it showed an even sharper upward trend with an AAPC of 15.8 (p<0.05). A study conducted in Uzbekistan in 2022 also showed a steady rise in obesity prevalence, but with a lower AAPC of 3.6 (p<0.001) [37].

The age-specific trends of overweight-obesity among both men and women showed the highest proportions in the 25–29 age group throughout the period, and the figures showed sharp upward trends. The prevalence of overweight-obesity among women in this age group increased from 2.2% in 1996 to 24.7% in 2019. This trend followed a similar pattern as that among women of reproductive age in Nepal, which showed a higher increase in the age group of 25 years and above [38]. We found that the men of the oldest (25–29 years) group had an overweight-obesity prevalence of 8.8% in 2007, which rose to 25.4% in 2019.

The prevalence of overweight and obesity significantly changed throughout the period from 1996 to 2019. Compared to the earliest survey in 1996, there were 9 times higher odds for overweight and 42 times higher odds for obesity in 2019, which shows an alarmingly rapid increase. Similarly, based on Asian cut-offs, the odds for being overweight among young adults were fourfold higher and the odds for being obese were 28 times greater compared to the survey in 1996. Similar to our findings, there was a substantial increase in the prevalence of overweight-obesity among Chinese adults [39]. The prevalence of overweight has doubled for Chinese women and tripled for men. This rapid increase in the trend might be attributed to the modernization and rapid change in lifestyle habits such as diet, physical activity, and use of tobacco and alcohol.

## Predictors/risk factors for overweight and obesity

We found that age, sex and marital status were strongly associated with overweight in a multivariable adjusted model. Similarly, age, sex and area of residence were significantly associated with obesity in a multivariable adjusted model. Young adults aged 20–24 years had twofold higher odds of being overweight, and those aged 25–29 years had fourfold higher odds of being overweight than those of less than 20 years. This showed an alarmingly increased risk in the older age groups, which calls for urgent intervention. This finding is supported by those of other cross-sectional studies conducted in Bangladesh, India and Guatemala, in which BMI, waist circumference and the waist to hip ratio were shown to increase with age [18, 40, 41]. Similarly, our study showed that the odds of being overweight and obese were 32% and 37% lower, respectively, among men than among women. The literature also shows that most developing countries are experiencing an increasing prevalence of overweight/obesity among women [20]. Consistent with our findings, in sub-Saharan Africa and Latin America, the likelihood of obesity in females is higher [36, 42]. However, male obesity was more prevalent than female obesity in Asian and North African countries [36]. This observation prompted an exploration of a hypothesis: whether social disadvantages for women result in reduced food intake by women and increased intake by men. Nevertheless, empirical verification of this hypothesis was inconclusive. In the NDHS 2016 survey, married adults had twice the odds of being overweight than their counterparts, similar to the findings of a study in Kuwait [43]. In the Kuwaiti population, being married was linked to higher odds of obesity/central obesity. We found that those residing in urban areas had twice the odds of being obese compared to

rural residents. This perhaps signifies that disadvantaged rural residents possibly contribute to a decreased prevalence of obesity. This was explained by another study in India in which BMI was significantly higher among urban dwellers than among those residing in rural regions [44]. The study showed a tenfold increase in the prevalence of overweight/obesity among urban young people. However, the result was the opposite in a study conducted in Uganda, which did not depict any difference in the prevalence of overweight between urban and rural dwellers [45].

## Strengths and limitations of this study

This study utilized national representative surveys, and thus, the results represented the national picture and can inform society and policy-makers on the increasing burden of overweight and obesity and provide guidance to plan for awareness programmes to help the young adults to keep themselves healthy. The use of two different surveys provided us with rich data from different years and enabled us to compare the findings of the two types of surveys. The DHSs are targeted to estimate the basic demographic and health indicators and focus mainly on general health issues including maternal and child health. On the other hand, the WHO STEPS surveys focus on assessing NCD risk factors among adults in Nepal. Therefore, including WHO survey data provided additional information to compare the status of overweight and obesity and the possible risk factors. The specific findings of one survey that it has ignored may be found in other surveys. Similarly, men were included only in the latest DHS; thus, including WHO surveys helped us compare the data and trends among men in different years.

However, some contrasting findings between surveys might create problems, resulting in questions regarding the reliability of the findings. For instance, the DHS data showed a higher prevalence of overweight among women, whereas the WHO data showed a higher prevalence among men. Such differences might be attributed to the variability in the sampling procedure and methodology. The WHO STEPS survey included highly selective samples and a relatively small sample size compared to the DHSs, which may have led to different findings. Moreover, detailed information such as physical activity, dietary habits, smoking and alcohol use were not available in all the data, which limited the assessment of the risk factors for overweight and obesity.

## Recommendations

Our study showed that overweight and obesity are the growing problems in Nepal that call for urgent obesity preventive measures through a programmed and integrated approach, such as early screening of obesity and possible risk factors, to mitigate present and future health risks and halt the rising obesity trend [3]. Our study results indicated that the upward trends in the prevalence of overweight and obesity were greatest in the 25–29 age group, which calls for urgent action towards the implementation of obesity control interventions targeting these risk groups at the national level. Young people need to reduce their calorie intake along with an emphasis on physical activity to reduce weight and cardiovascular risk [46]. If unchecked, more of these productive groups will tend to be overweight or obese, which ultimately leads to increasing morbidity and a higher burden on society. Policy-makers, health workers and the community at large must join forces in addressing this global public health problem [3], specifically in the high-risk groups.

The study findings will be shared with the concerned stakeholders and policy-makers for the codevelopment of intervention strategies to prevent overweight and obesity.

## Conclusions

We found a significant and rapid increase in trends of overweight and obesity for both men and women, with a slight drop in 2007. The odds of overweight increased ninefold in 2019 compared to the earliest survey, whereas the odds of obesity increased 42-fold over the same period. Based on a survey conducted in 2016, older age groups and women had higher odds of being overweight and obese than younger age groups and men. Being married was a strong predictor for overweight, while residing in an urban area posed a higher risk for obesity.

## Supporting information

**S1 Fig.** Trends of overweight and obesity among women and men from 1996 to 2019 using the WHO cutoffs and **(b)** Asian cutoffs.
(DOCX)

**S2 Fig. Trends of median waist-to-hip ratio and waist-to-height ratio.**
(DOCX)

**S1 Table. Demographic characteristics of women by years of survey.**
(DOCX)

**S2 Table. Demographic characteristics of men by years of survey.**
(DOCX)

**S3 Table. STROBE checklist.**
(DOCX)

## Acknowledgments

We are also thankful to the DHS program and WHO NCD microdata repository for allowing us to access the data of the NDHS survey and WHO STEPS survey. Lastly, we would like to express our gratitude to our colleague, Amado Quezada Sanchez, who helped in data analysis and writing.

## Author Contributions

**Conceptualization:** Sujata Shakya, Pilvikki Absetz, Subas Neupane.

**Data curation:** Sujata Shakya, Subas Neupane.

**Formal analysis:** Sujata Shakya, Subas Neupane.

**Methodology:** Sujata Shakya, Pilvikki Absetz, Subas Neupane.

**Resources:** Sujata Shakya.

**Software:** Sujata Shakya, Subas Neupane.

**Supervision:** Pilvikki Absetz, Subas Neupane.

**Validation:** Sujata Shakya, Subas Neupane.

**Visualization:** Sujata Shakya.

**Writing – original draft:** Sujata Shakya, Pilvikki Absetz, Subas Neupane.

**Writing – review & editing:** Sujata Shakya, Pilvikki Absetz, Subas Neupane.

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
