## [Decision Letter · Decision Letter 0]

27 Jun 2023

PGPH-D-23-00683

Two-decade trends and factors associated with overweight and obesity among young adults in Nepal

Dear Dr. Shakya,

Thank you for submitting your manuscript to PLOS Global Public Health. After careful consideration, we feel that it has merit but does not fully meet PLOS Global Public Health’s publication criteria as it currently stands. Therefore, we invite you to submit a revised version of the manuscript that addresses the points raised during the review process.

EDITOR:

Dear Author,

This manuscript still requires major corrections. Please attend to all the comments provided by the reviewer/s.

The decision of this manuscript is justified based on PLOS Global Public Health’s publication criteria and not on its novelty or perceived impact.

We look forward to receiving your revised manuscript.

Kind regards,

Zulkarnain Jaafar

Academic Editor

Journal Requirements:

1. Please upload a copy of Figure 3b which you refer to in your text on page 10. Or, if the figure is no longer to be included as part of the submission please remove all reference to it within the text.

2. In the online submission form, you indicated that "Data are already open for public use upon permission". All PLOS journals now require all data underlying the findings described in their manuscript to be freely available to other researchers, either 1. In a public repository, 2. Within the manuscript itself, or 3. Uploaded as supplementary information.

Additional Editor Comments (if provided):

Reviewers' comments:

Reviewer's Responses to Questions

**Comments to the Author**

1. Does this manuscript meet PLOS Global Public Health’s publication criteria? Is the manuscript technically sound, and do the data support the conclusions? The manuscript must describe methodologically and ethically rigorous research with conclusions that are appropriately drawn based on the data presented.

Reviewer #1: Yes

Reviewer #2: Yes

2. Has the statistical analysis been performed appropriately and rigorously?

Reviewer #1: Yes

Reviewer #2: I don't know

3. Have the authors made all data underlying the findings in their manuscript fully available (please refer to the Data Availability Statement at the start of the manuscript PDF file)?

Reviewer #1: Yes

Reviewer #2: Yes

4. Is the manuscript presented in an intelligible fashion and written in standard English?

Reviewer #1: Yes

Reviewer #2: No

5. Review Comments to the Author

Reviewer #1: thanks for invitation of this good manuscript

the article is well written with good analysis and the topic is important , please you can only add paragraph for recommendation based on you findings' and dissemination plan of your results

Reviewer #2: Below are my main comments:

-general comment: many weak sentence structures; avoid long sentences,

- it was mentioned that "Young adulthood of age 18 to 25 years, is a critical stage.." but it is not explained why you included the target group till 29 years old?

- some sentences are missing in- text citation.

- It was mentioned "There are inadequate evidences in Nepal to explain the current trends of overweight and obesity among young adults group and the socio-economic factors that are underway. There are only scattered findings from small scale studies through which it is challenging to draw concrete conclusion." but you already mentioned above that "Different studies in Nepal showed high prevalence among the young adults group [15,17,25]". Contradictory statements.

The study design is not clear enough. It was mentioned " We used several cross-sectional surveys.." Do you mean you used the data collected previously from these surveys? You did not do primary data but used data collected previously?

- "The total participants enrolling in the surveys in different years were 21,281." Is this number related to the age group 18-29 only?

- in the WHO STEPS Surveys: "The STEPS surveys are population-based surveys among adults aged 15–69 years conducted using internationally comparable, standardized and integrated surveillance tool.." but you took data from the age group 18-29 only? please explain

- in the DHS surveys you stated that “We utilized data from five surveys between 1996 and 2016.” was it continuously? or for a few years only? please explain

-What type of data did you collect from each of these surveys? This should be clearly stated.

- ethical consideration: did you take IRB? Did you use identifiable data from the two surveys?

- it was stated “We found very limited studies on trends of overweight/obesity prevalence, especially among the young adults to compare with our findings, as most of the studies were conducted among reproductive aged (15- 49 years) women.” But your age group (18-29) is included in this group so why is this a limitation?

- it was stated, “The DHS surveys are targeted to estimate the basic demographic and health indicators and focus mainly on maternal and child health issues.” How is maternal and child health related to your research and target group (18-29)?

- “Our study results indicate that the highest risk groups for increasing overweight and obesity prevalence are the age-group 25-29 years,” how did you conclude this?? Among which risk groups?

6. PLOS authors have the option to publish the peer review history of their article (what does this mean?). If published, this will include your full peer review and any attached files.

**Do you want your identity to be public for this peer review?** For information about this choice, including consent withdrawal, please see our Privacy Policy.

Reviewer #1: **Yes: **Wafaa Yousif Abdel-Wahed

Reviewer #2: No

---

## [Decision Letter · Decision Letter 1]

19 Sep 2023

PGPH-D-23-00683R1

Two-decade trends and factors associated with overweight and obesity among young adults in Nepal

Dear Dr. Shakya,

Thank you for submitting your manuscript to PLOS Global Public Health. After careful consideration, we feel that it has merit but does not fully meet PLOS Global Public Health’s publication criteria as it currently stands. Therefore, we invite you to submit a revised version of the manuscript that addresses the points raised during the review process.

EDITOR: Dear author,

There is still some minor revisions needed to be made in this manuscript. Please attend to the reviewer comments to improve your manuscript further.

The decision of this manuscript is justified based on PLOS Global Public Health’s publication criteria and not for example on its novelty or perceived impact.

We look forward to receiving your revised manuscript.

Kind regards,

Zulkarnain Jaafar

Academic Editor

Journal Requirements:

Additional Editor Comments (if provided):

Reviewers' comments:

Reviewer's Responses to Questions

**Comments to the Author**

1. If the authors have adequately addressed your comments raised in a previous round of review and you feel that this manuscript is now acceptable for publication, you may indicate that here to bypass the “Comments to the Author” section, enter your conflict of interest statement in the “Confidential to Editor” section, and submit your "Accept" recommendation.

Reviewer #3: All comments have been addressed

2. Does this manuscript meet PLOS Global Public Health’s publication criteria? Is the manuscript technically sound, and do the data support the conclusions? The manuscript must describe methodologically and ethically rigorous research with conclusions that are appropriately drawn based on the data presented.

Reviewer #3: Yes

3. Has the statistical analysis been performed appropriately and rigorously?

Reviewer #3: Yes

4. Have the authors made all data underlying the findings in their manuscript fully available (please refer to the Data Availability Statement at the start of the manuscript PDF file)?

Reviewer #3: Yes

5. Is the manuscript presented in an intelligible fashion and written in standard English?

Reviewer #3: Yes

6. Review Comments to the Author

Reviewer #3: This manuscript presents an important investigation of overweight and obesity in youth, measured over two decades period. The combination of information sources and the application of various analysis methodologies constitute good support for the information obtained. The analysis is complete and indicates the scope and limitations of the data of the trends described as a result of the research. The discussion of the results is pertinent and is aimed at decision makers regarding this public health problem. The conclusions are correctly written and are consistent with the objectives. However, I would like to make two writing suggestions that may improve it:

In section Discussion

Sentence: "Older age, female sex and married status increased the likelihood of overweight, whereas older age, female sex and urban residence increased the likelihood for obesity."

I suggest: The likelihood of overweight increased in groups of older age, female sex and married status, whereas the likelihood of obesity was increased in groups of older age, female sex and urban residence.

In section Predictors/risk factors for overweight and obesity

Sentence: "However, male obesity was found to be more common than female obesity in Asian and North African countries [37]. This was linked to the social disadvantages for women leading to less food intake by women and more intake by men; however, this was not empirically confirmed."

I suggest: However, male obesity was found to be more common than female obesity in Asian and North African countries [37]. That finding wanted to be explained with the hypothesis: the social disadvantages for women leading to less food intake by women and more intake by men; however, it could not be verified empirically.

Edison Aguilar Santacruz

7. PLOS authors have the option to publish the peer review history of their article (what does this mean?). If published, this will include your full peer review and any attached files.

**Do you want your identity to be public for this peer review?** For information about this choice, including consent withdrawal, please see our Privacy Policy.

Reviewer #3: **Yes: **Edison Aguilar Santacruz

---

## [Editor Report · Decision Letter 2]

28 Sep 2023

PGPH-D-23-00683R2

Two-decade trends and factors associated with overweight and obesity among young adults in Nepal

Dear Dr. Shakya,

Thank you for submitting your manuscript to PLOS Global Public Health. After careful consideration, we feel that it has merit but does not fully meet PLOS Global Public Health’s publication criteria as it currently stands. Therefore, we invite you to submit a revised version of the manuscript that addresses the points raised during the review process.

EDITOR: Dear author,

There are still some minor revisions that need to be made to this manuscript. Please attend to the reviewer's comments to improve your manuscript further.

The decision of this manuscript is justified based on PLOS Global Public Health’s publication criteria and not for example on its novelty or perceived impact.

We look forward to receiving your revised manuscript.

Kind regards,

Zulkarnain Jaafar

Academic Editor
---

## [Editor Report · Decision Letter 3]

4 Oct 2023

Two-decade trends and factors associated with overweight and obesity among young adults in Nepal

PGPH-D-23-00683R3

Dear Dr. Shakya,

We are pleased to inform you that your manuscript 'Two-decade trends and factors associated with overweight and obesity among young adults in Nepal' has been provisionally accepted for publication in PLOS Global Public Health.

Best regards,

Zulkarnain Jaafar

Academic Editor